# Evaporation Driven Hydrovoltaic Generator Based on Nano-Alumina-Coated Polyethylene Terephthalate Film

**DOI:** 10.3390/polym15204079

**Published:** 2023-10-13

**Authors:** Shipu Jiao, Yihao Zhang, Yang Li, Bushra Maryam, Shuo Xu, Wanxin Liu, Miao Liu, Jiaxuan Li, Xu Zhang, Xianhua Liu

**Affiliations:** School of Environmental Science and Engineering, Tianjin University, Tianjin 300354, China; jsp@tju.edu.cn (S.J.); zhangyihao_@tju.edu.cn (Y.Z.); liyang_@tju.edu.cn (Y.L.); maryambushra@yahoo.com (B.M.); xushuo001226@163.com (S.X.); liuwanxin1125@163.com (W.L.); liumiao@tju.edu.cn (M.L.); ljxx@tju.edu.cn (J.L.); zhangxu_2022@tju.edu.cn (X.Z.)

**Keywords:** Al_2_O_3_, polyethylene terephthalate film, evaporation, electric double layer, hydrovoltaic effect

## Abstract

Collecting energy from the ambient environment through green and sustainable methods is highly expected to alleviate pollution and energy problems worldwide. Here, we report a facile and flexible hydrovoltaic generator capable of utilizing natural water evaporation for sustainable electricity production. The generator was fabricated by coating nano-Al_2_O_3_ on a twistable polyethylene terephthalate film. An open circuit voltage of 1.7 V was obtained on a piece of centimeter-sized hydrovoltaic generator under ambient conditions. The supercapacitor charged by the hydrovoltaic device can power a mini-motor efficiently. Moreover, by expanding the size or connecting it in series/parallel, the energy output of the generator can be further improved. Finally, the influence factors and the mechanism for power generation were primarily investigated. Electrical energy is produced by the migration of water through charged capillary channels. The environmental conditions, the properties of the solution and the morphology of the film have important effects on the electrical performance. This study is anticipated to offer enlightenment into designing novel hydrovoltaic devices, providing diverse energy sources for various self-powered devices and systems.

## 1. Introduction

Water is the most abundant and common liquid on Earth [1,2,3]. Due to the spontaneity and universality of water evaporation, energy harvesting devices that use water as energy have received great attention [4,5,6,7,8,9,10,11]. In 1859, Quincke discovered a potential difference at the ends of a pumping pipe [12]. In 2003, Ghosh et al. reported the generation of an output voltage of 0.65 mV when pure water flowed through a single-walled carbon nanotube bundle [13]. Until 2017, Zhou et al. generated a sustained Voc of up to 1 V by evaporating water from a porous carbon black membrane [14]. After nearly two decades of development, the performance of the generator was improved by three orders of magnitude. Since then, researchers have developed a large number of hydrovoltaic materials, including carbon materials, metal oxides, biofibers, and polymers [15,16,17,18,19]. The power generation in these devices stems from the water–solid interactions and is primarily based on ionic diffusion or directional migration of water in charged micro-nano channels to generate current. The surface charge of the channel has an important effect on the polarity and size of the electrical energy. In addition, materials with good hydrophilicity and high specific surface area are suitable for hydrovoltaic power generation. Because the generators can generate electricity from widespread moisture and water evaporation, hydrovoltaic technology has a bright future in new self-powered, battery-free monitoring/diagnostic systems in healthcare, the Internet of things, the environment, and artificial intelligence [20]. However, there are still some problems in the study of hydrovoltaic power generation, such as controversy regarding the mechanism of the hydrovoltaic effect, the poor power generation performance of the equipment, and the long-term condition and stability of the equipment that require further improvements [6].

Alumina is a type of compound composed of aluminum, oxygen, and usually hydrogen. It is a white powdery solid, which is non-toxic and odorless [21]. As an inorganic non-metallic powder material, alumina has excellent properties such as high strength, high hardness, wear resistance, corrosion resistance, high-temperature resistance, oxidation resistance, and a large surface area [22,23]. It is an important chemical material with high market demand [24]. With the refinement of particles, the electronic and crystal structures on its surface have undergone changes, resulting in surface effects, small size effects, quantum effects, and macroscopic quantum tunneling effects that macroscopic objects do not possess. This has enabled alumina powder to exhibit a series of excellent properties in many electrical, optical, mechanical, and chemical reactions [25,26,27]. Alumina has a wide range of applications in semiconductor materials, surface protective materials, electronic components, optical materials, biomedical materials, and other fields due to its excellent performance [28,29,30,31].

In this report, by using a simple process of scraping and coating alumina dispersion on a polyethylene terephthalate (PET) film at room temperature, we fabricated a flexible photovoltaic generator with a continuous output of electrical energy driven by water evaporation. Under controlled environmental conditions, a single generator generated a Voc of 1.7 V and can be used to charge the supercapacitor to drive the micro-motor to operate normally. In addition, the power generation mechanism and influencing factors of the device’s performance were studied and discussed. Furthermore, the energy output of a hydrovoltaic generator can be further boosted by expanding its size or connecting it in series/parallel. This hydrovoltaic technology is simple, green, and convenient-to-operate. This report is expected to expand the range of materials suitable for evaporative drive generators, providing new ideas for the construction of novel efficient energy conversion devices.

## 2. Materials and Methods

### 2.1. Material

Al_2_O_3_ powders (99.99%, molecular weight = 101.96, α-phase) with different diameters (30 nm, 50 nm, 250 nm and 1 micron) were purchased from Yongye wear-resistant material company (Yongzhou, China); Conductive carbon paste (CH-8) was purchased from Jelcon (Tokyo, Japan); PET film was purchased from Renyuan (Shanhai, China). All the materials were used without any further purification.

### 2.2. Production of the Hydrovoltaic Generator

PET films were cleaned in ethanol by ultrasonication, and, after drying naturally, conductive carbon paste was coated on PET as an L-shaped electrode. The distance between the top electrode and the bottom electrode was 1.5 cm, and the line width of the electrode was 1 cm. In addition to studying the effect of different particle size nanoparticles on the performance of a hydrovoltaic generator, the size of Al_2_O_3_ nanoparticles in the experimental setup was 250 nm. The Al_2_O_3_ was dispersed in ethanol and sonicated for 4 h to obtain α-Al_2_O_3_ slurry. Then, the α-Al_2_O_3_ slurry was evenly scraped onto the PET substrate with electrodes using a film scraper. As the ethanol evaporates, the α-Al_2_O_3_ slurry began to shrink. Under the action of capillary force, the nanoparticles were close to each other, and they finally formed a solid porous structure, forming the skeleton of the dry film. Until the film was dry, the nanoparticles were tightly combined with the matrix, and a porous α-Al_2_O_3_ slurry functional layer with abundant nanochannels was obtained.

### 2.3. Characterization

The Voc and Isc of the hydrovoltaic generator were measured by using a CHI-1400 electrochemical workstation (Chenhua, Shanhai, China). All experiments were measured using carbon electrodes. The resistance was measured with a multimeter (victor 8145B). All experiments were performed in the laboratory with a relative humidity of 20% ± 5 and a temperature of 23 °C ± 2. A S-4800 scanning electron microscopy (SEM) (Hitachi, Tokyo, Japan) was used to obtain scanning electron microscopy images of the sample. The zeta potential was measured using a nanoparticle potential analyzer (Zetasizer Nano ZS, Marvern, UK).

## 3. Results and Discussion

### 3.1. The Preparation of the Hydrovoltaic Generator

Appendix A shows the schematic diagram of the preparation process of the hydrovoltaic generator. Initially, a film-scraping machine was used to coat the PET film with commercial conductive carbon paste as the electrode. Then, after the carbon paste dried, Al_2_O_3_ nanoparticle dispersion (0.5 g mL^−1^, ethanol) was applied between the electrodes. Al_2_O_3_ nanoparticles have good dispersion (Appendix A). The prepared hydrovoltaic generator is shown in Appendix A. The Al_2_O_3_-based hydrovoltaic generator has good flexibility and can still return to its original state after multiple bendings (Appendix A). The Al_2_O_3_ particles adhere tightly to the substrate due to capillary-driven self-assembly during the dispersion liquid drying process; in addition, van der Waals forces and hydrogen bonds also contribute to the adhesion between the nanoparticles and between the substrate and the nanoparticles [32].

### 3.2. Electricity Generation from Evaporation

A long-term real-time Voc test was conducted to verify the power generation performance of the Al_2_O_3_-based hydrovoltaic generator. By soaking the bottom of the generator in deionized water, a Voc of 1.7 V was generated at a natural room temperature (23 °C, 20% RH) (Figure 1a,b). The morphology and zeta potential of the Al_2_O_3_ film were studied. As shown in Figure 1c, Al_2_O_3_ thin film was composed of porous nanochannels, which provided a path for water migration. The zeta potential of Al_2_O_3_ nanoparticles is as high as +30 mV, and the Al_2_O_3_ nanochannel is positively charged. Driven by capillary force, water passes upward through the nanochannel, and a balance between water evaporation and capillary permeation flux is finally formed due to the evaporation of water on the membrane surface (Figure 1d,e, Appendix A). During water flow, the positively charged nanochannel repels positively charged ions (H^+^), but it allows negatively charged hydroxyl ions (OH^−^) to pass through, resulting in a current potential and charge accumulation in the water flow, and finally the device reaches a state where the stable Voc is numerically stable and the top charge is extremely negative (Figure 1f) [33].

To further demonstrate that the manufactured hydrovoltaic generator was driven primarily by evaporation, the device was sealed in a glass container (Figure 2a). As the water vapor gradually reached the saturation point, the water evaporation on the surface of the Al_2_O_3_ film gradually stopped, the induced voltage correspondingly dropped, and finally disappeared within 2000 s. After opening, the voltage quickly recovered to 1.7 V, and the voltage changes caused by sealing and unsealing could be completely repeated (Figure 2b). The voltage variation indicates that water evaporation is essential for continuous power generation of a water-volt generator [34]. We further carried out experiments to verify that when the ambient temperature increased from 10 °C to 40 °C, the Voc increased from 1.2 V to 2.3 V, and the Isc increased from 0.1 μA to 0.22 μA (Figure 2c,d), mainly because the high temperature accelerated the evaporation of the water.

To determine the contribution of the evaporation-driven flow potential in generating electrical energy in the Al_2_O_3_-based hydrovoltaic device, a specific four-electrode experiment was designed with the four electrodes (referred to as E1–E4) spaced on average about 0.5 cm apart (Figure 2e). One bottom electrode (E1) of the four-electrode device was inserted into the water, while the remaining electrodes (E2, E3, E4) remained above the water level. Under the action of capillary forces and water evaporation, the water gradually submerged all electrodes from bottom to top.

The Voc (V_1-2_, V_1-3_, V_1-4_) between E1–E2, E1–E3, and E1–E4 was measured, and it was found that the Vocs of V_1-2_, V_1-3_, V_1-4_ were 0.55 V, 0.72 V, and 1.68 V, respectively (Figure 2f). This indicated that the Al_2_O_3_ film in each section contributed to the voltage generation. The source of the Voc was due to the flow potential. Similarly, the Isc (I_1-2_, I_1-3_, I_1-4_) between E1–E2, E1–E3, and E1–E4 was measured. I_1-2_ basically generated no current, I_1-3_ generated a weak current (0.02 μA), and I_1-4_ generated an Isc of 0.15 μA (Figure 2g). This showed that the causes of the Voc and Isc were different, and it showed that the Isc is mainly caused by the water evaporation interface of the upper Al_2_O_3_ film [35].

### 3.3. Influence Factors

In order to understand the effect of Al_2_O_3_ film size and Al_2_O_3_ nanoparticle size on power generation, further detailed experimental evidence was obtained. Firstly, the influence of film length on power generation was studied. When the film length was increased from 0.5 cm to 1.5 cm, the Voc was increased from 0.55 V to 1.7 V (Figure 3a), while the Isc remained basically unchanged (about 0.15 microamps) (Figure 3b). The Voc was positively correlated with the length of the film, while the Isc was independent of the length of the film. Next, the influence of the film width was studied. When the film width was increased from 2 cm to 4 cm, the Voc basically did not change (about 1.7 V) (Figure 3c), and the Isc increased from 0.12 μA to 0.18 μA (Figure 3d). This showed that the Voc of the generator was independent of the width of the film, while the Isc was positively correlated with the width of the film. This phenomenon can be explained well according to the expression of current and Ohm’s law.

Current expression:(1)I=nqvS
where *n* represents the number of charged particles per unit volume, *q* represents the amount of charge of a single particle, *v* represents the speed at which the charged particles move, and *S* represents the cross-sectional area.

Resistance law:(2)R=ρL/S
where *ρ* is the resistivity, *L* is the resistance length, and *S* is the resistance cross-section area

Ohm’s law:(3)U=IR

The *I*, *U*, and *R* are the current intensity, voltage, and resistance belonging to the same part of the circuit at the same time.

By combining Equations (1)–(3), we can obtain:(4)U=nqvρL

It can be seen from Equation (1) that increasing the width of the film increases the cross-sectional area, which is equivalent to multiple batteries in parallel, thereby increasing the Isc. It can be seen from Equation (4) that increasing the length of the film is equivalent to multiple batteries in series, thus increasing the Voc, and the Voc and the Isc are related to the movement speed of the charged particles (*v*). When favorable and evaporating conditions are created, the movement speed of charged particles can be improved, thus improving the output performance of the water-volt generator. This conclusion agrees with the results of the evaporation experiment.

The Al_2_O_3_ particle size will affect the migration of water, thus affecting the performance of the hydrovoltaic generator. As can be seen from Figure 3e,f, when the particle size was increased from 30 nm to 250 nm, the Voc was increased from 1.2 V to 1.7 V, and the Isc was increased from 0.06 μA to 0.15 μA. When the particle size was 1 μm, the Voc and Isc dropped to 0.2 V and 0.015 μA, respectively. This was because when the particle size was too small, it caused the narrow water channel, reduced the movement speed of charged particles, resulting in a relatively small output performance, and when the particle size was too large, it led to a decline in the screening of counter-ions, and the number of charged particles was reduced, resulting in lower electricity generation performance. It is clear that the hydrovoltaic generator’s output performance is significantly influenced by the particle size.

In order to further study the factors affecting the performance of the Al_2_O_3_-based hydrovoltaic generator, we carried out evaporation-induced potential experiments with salt solutions containing different concentrations of NaCl. The results showed that, with the increase in NaCl concentration from 10^−6^ mol L^−1^ to 0.1 mol L^−1^, the Voc and the Isc decreased significantly. The Voc decreased from 1.69 V to 0.08 V (Figure 4a), and the Isc decreased from 0.15 μA to 0.007 μA (Figure 4b).

Similar results were obtained with different pH solutions. When the solution was neutral, the hydrovoltaic generator achieved the maximum output performance, with Voc and Isc of 1.7 V and 0.15 μA, respectively. With the increase of the ion content in the solution (pH becomes 1 or 13), the Voc and the Isc of the hydrovoltaic generator decreased (Figure 4c,d).
(5)$$\lambda D=\sqrt{\varepsilon \varepsilon 0kBT/2nbulkz^{2}e^{2}}$$
where *ε* is the dielectric constant of the medium and *ε_0_* is the dielectric coefficient of the vacuum. k_B_ is the Boltzmann constant, *T* is the absolute temperature, *n_bulk_* is the concentration of the bulk ions, *z* is the valence state of the ions, and e is the charge of the electrons. According to Equation (5), with the increase in ion concentration, Debye length (*λ_D_*) will become shorter. The *λ_D_* of the solution with a higher ion concentration is shorter than that in deionized water. This leads to a small EDL effect [36]. Therefore, when the solution is neutral, the hydrovoltaic generator has better output performance, and the high ion concentration of the solution will make the Voc and the Isc of the hydrovoltaic generator decrease significantly.

### 3.4. Application of the Device

The output voltage increased with the increase in the resistance, and the output current decreased with the increase in the resistance. When the external resistance was 10 MΩ, it was equal to the internal resistance of the hydrovoltaic generator, and the maximum output voltage was 0.068 μW (Figure 5a). Since the output current of the generator is small, it is difficult to directly use the generated electrical energy. The supercapacitor of 5 F can be charged by using the hydrovoltaic generator (Figure 5b). After 10 h of charging, the micro-motor can be driven for 20 s (Appendix A). The output performance can be further amplified by connecting the hydrovoltaic generators in series and in parallel. By connecting eight generators in series, a Voc of up to 12 V can be obtained (Figure 5c), and by connecting eight generators in parallel, an Isc of 1 μA can be obtained (Figure 5d). Our work therefore provides a new, green way to generate electricity by exploiting cheap Al_2_O_3_ particles and the ubiquitous phenomenon of evaporation.

## 4. Conclusions

In short, the simple and fast coating technology creates an Al_2_O_3_-based hydrovoltaic generator. Under optimal environmental conditions, the water evaporated in the centimeter-sized Al_2_O_3_ film, which is only a few microns thick, can generate 1.7 V of high pressure. Compared with other methods, hydrovoltaic technology does not require energy from complex environments, such as solar wind and tidal energy. There is also no need for external energy supplies, such as pressure gradients, temperature gradients, and chemical concentration gradients. The hydrovoltaic generator continuously converts the ambient low-grade heat energy into electricity by utilizing the spontaneous evaporation of water everywhere in nature. By increasing the size or series/parallel connection, the hydrovoltaic generator’s energy output can be further improved to power numerous electronic components. Hydrovoltaic technology offers new concepts and a wide range of development space for the creation of new efficient energy conversion devices.

## Figures and Tables

**Figure 1 polymers-15-04079-f001:**
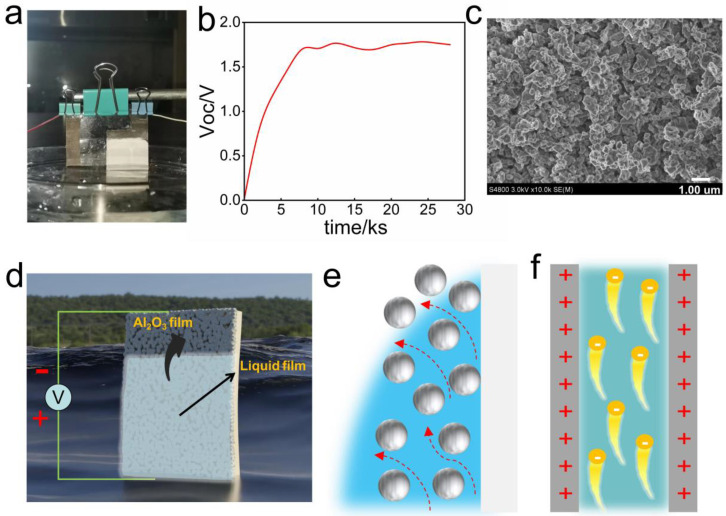
Power generation of the Al_2_O_3_-based hydrovoltaic generator driven by water evaporation. (**a**) Photo of the Al_2_O_3_-based hydrovoltaic generator; (**b**) Voc of the hydrovoltaic generator under environmental conditions; (**c**) scanning electron microscopy images of the Al_2_O_3_ film; (**d**) structural diagram of the Al_2_O_3_-based hydrovoltaic generator; (**e**) schematic diagram of water migration in the nanochannels of the Al_2_O_3_ film (the arrow represents the direction of water migration); (**f**) schematic diagram of the electricity generation in the Al_2_O_3_-based hydrovoltaic generator (the microchannel is positively charged).

**Figure 2 polymers-15-04079-f002:**
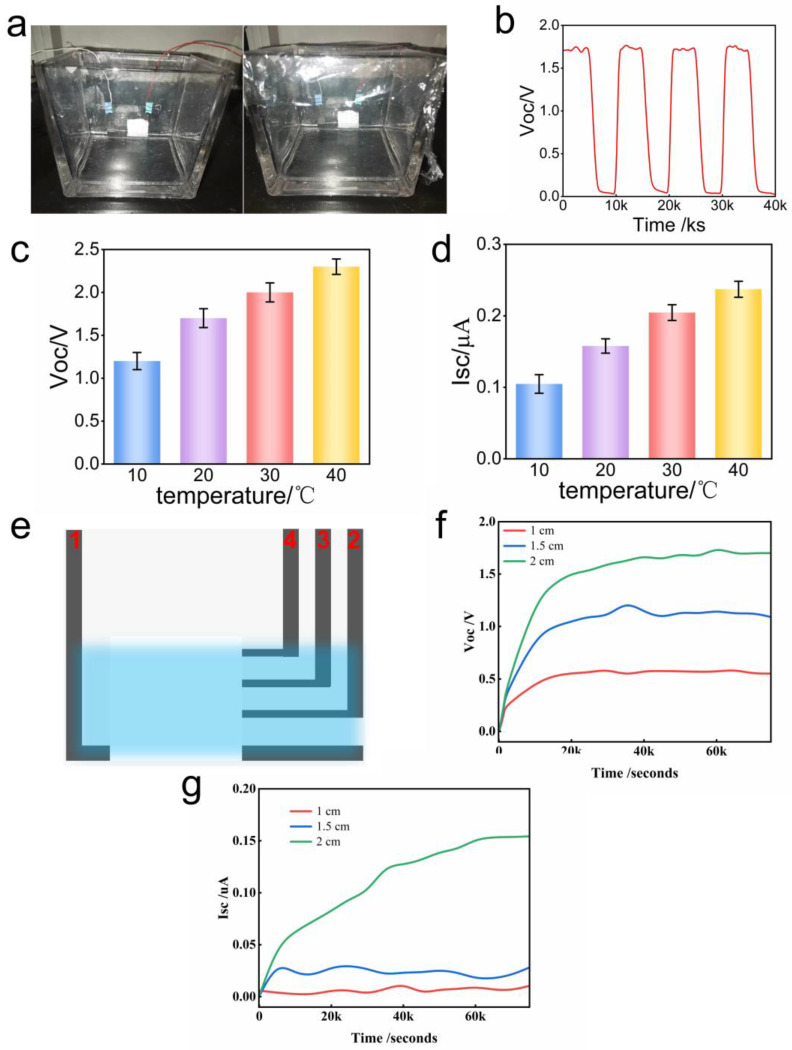
Evaporation experiment of the hydrovoltaic generator. (**a**) Photos of the hydrovoltaic generators in open and sealed states; (**b**) the Voc of the hydrovoltaic generator in open and sealed states; (**c**) Voc of the hydrovoltaic generators at different temperatures; (**d**) Isc of the hydrovoltaic generators at different temperatures; (**e**) schematic diagram of the four-electrode experimental device (the number represents the electrode number); (**f**) Voc of V_1-2_, V_1-3_, V_1-4_. (**g**) Isc of I_1-2_, I_1-3_, I_1-4_.

**Figure 3 polymers-15-04079-f003:**
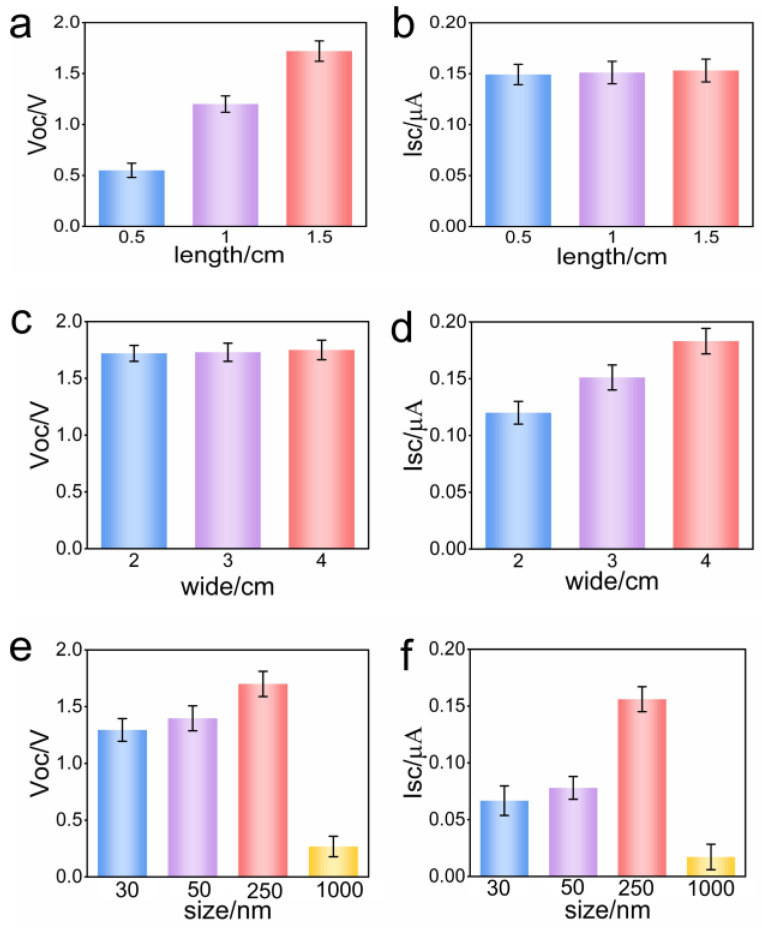
The influence of the Al_2_O_3_ film size and the Al_2_O_3_ nanoparticle size on the power generation performance of the hydrovoltaic generator. (**a**) Voc of Al_2_O_3_ films with different lengths; (**b**) Isc of Al_2_O_3_ films with different lengths; (**c**) Voc of Al_2_O_3_ films with different widths; (**d**) Isc of Al_2_O_3_ films with different widths; (**e**) Voc of Al_2_O_3_ films with different particle sizes; (**f**) Isc of Al_2_O_3_ films with different particle sizes.

**Figure 4 polymers-15-04079-f004:**
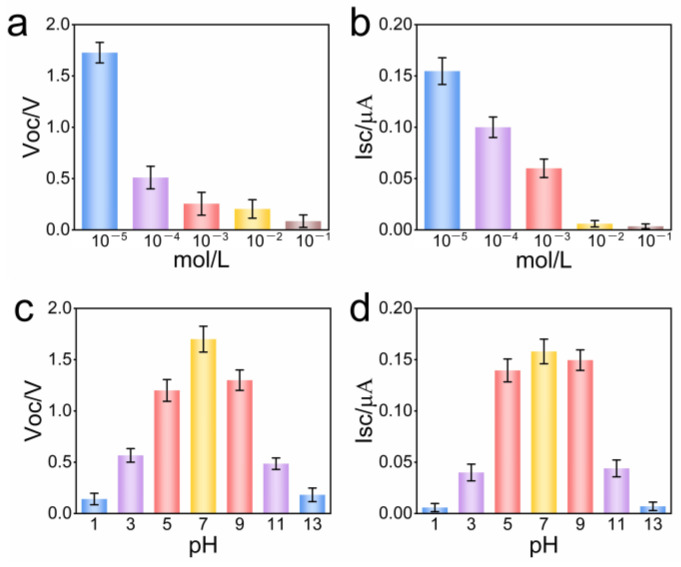
The influence of different solutions on the power generation performance of the hydrovoltaic generator. (**a**) Voc of NaCl solutions with different concentrations; (**b**) Isc of NaCl solutions with different concentrations; (**c**) Voc of different pH solutions; (**d**) Isc of different pH solutions.

**Figure 5 polymers-15-04079-f005:**
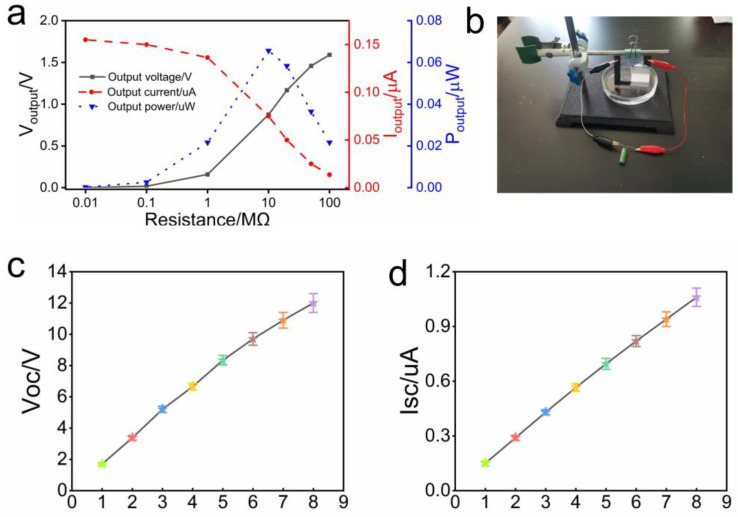
Application and scaling up of the Al_2_O_3_-based hydrovoltaic generator. (**a**) Output voltage, output current, and output power of a single hydrovoltaic generator under different loads; (**b**) photo of a single hydrovoltaic generator charging supercapacitors; (**c**) Voc of multiple hydrovoltaic generators connected in series (1–8); (**d**) Isc of multiple hydrovoltaic generators in parallel.

## Data Availability

Not applicable.

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
