# Peer review of "Evaporation Driven Hydrovoltaic Generator Based on Nano-Alumina-Coated Polyethylene Terephthalate Film"

_polymers, 2023, doi:10.3390/polym15204079_

Round 1

Reviewer 1 Report

It is an interesting work of the authors, who have reported a facile and flexible hydrovoltaic generator capable of utilizing natural water evaporation for sustainable electricity production. I think the langunage needs a minior correction, even though the basic science discussed is adequate. This work therefore may be recommended for publication after a minor revision suggested.

--

Author Response

Comment: It is an interesting work of the authors, who have reported a facile and flexible hydrovoltaic generator capable of utilizing natural water evaporation for sustainable electricity production. I think the language needs a minor correction, even though the basic science discussed is adequate. This work therefore may be recommended for publication after a minor revision suggested.

Response: Many thanks for the positive comment. The revised manuscript was checked and edited carefully. Major changes are marked in RED in the revised manuscript.

Reviewer 2 Report

Manuscript entitledEvaporation Driven Hydrovoltaic Generator Based on Nano-alumina Coated Polyethylene Terephthalate Filmsubmitted by Shipu Jiao, Yihao Zhang, Yang Li, Bushra Maryam, Shuo Xu, Wanxin Liu, Miao Liu, Jiaxuan Li, Xu Zhang and Xianhua Liu, can be considered for publication in Polymers Journal, after a major revision.

Here is a list of my specific comments:

1.     Page 1, Abstract: Include in this section the most important experimental results to highlight the importance of this study.

2.     Page 1, 1. Introduction: This section is too brief and should be detailed. The most important aspects related to this topic should be clearly presented in order to provide a properly presentation of the state of art in this field. The mist important drawbacks of the current approaches should be emphasized.

3.     Page 1, lines 28-45: “Especially in 2017, Zhou…”.These examples should be systematized. Avoid presenting each study from the literature.

4.     Page 2, 2.2. Production of hydrovoltaic generator: In this section, the most important technical details should be added.

5.     Page 3, Figure 1: This figure is too simple and should be moved into Supplementary materials.

6.     Page 8, 4. Application: Include this section in Results and discussions.

Author Response

Comment 1: Page 1, Abstract: Include in this section the most important experimental results to highlight the importance of this study.

Response: Thank you for your suggestion. We have revised the abstract to include important experimental results.

“Electrical energy is produced by the migration of water through charged capillary channels. The environmental conditions, the properties of the solution and the morphology of the film have important effects on the electrical performance.”

Comment 2: Page 1, 1. Introduction: This section is too brief and should be detailed. The most important aspects related to this topic should be clearly presented in order to provide a properly presentation of the state of art in this field. The mist important drawbacks of the current approaches should be emphasized.

Response: We have made changes in the introduction to provide a properly presentation of the state of art in this field and emphasized the important drawbacks of the current approaches.

“The power generation in these devices comes from the water-solid interactions and is primarily based on ionic diffusion or directional migration of water in charged micro/nanochannels. Good hydrophilicity, suitable size of nanochannels and high surface area of materials are important characteristics for excellent hydrovoltaic power generation [5,6,11]. In addition, the surface charge density and zeta potential of the channels have significant impacts on the electrical polarity and output performance of the hydrovoltaic devices [5,6]. Hydrovoltaic generators can generate electricity from widespread moisture and water evaporation. Due to its advantages of low cost, no pollution, and high portability, hydrovoltaic technology has a bright future in the fields of self-powered monitoring/diagnostic systems, the Internet of Things, and artificial intelligence [13]. However, its practical application still faces some bottlenecks, such as the working mechanisms need to be further clarified, the output power of the devices is low, and the long-term stability need to be further improved [6].”

Comment 3: Page 1, lines 28-45: “Especially in 2017, Zhou…”. These examples should be systematized. Avoid presenting each study from the literature.

Response: Thank you for the suggestion, we have systematized these examples.

“Until 2017, Zhou et al. generated a sustained Voc of up to 1 V by evaporating water from a porous carbon black membrane [14]. After nearly two decades of development, the performance of the generator was improved by three orders of magnitude. Since then, researchers have developed a large number of hydrovoltaic materials, including carbon materials, metal oxides, biofibers, and polymers [15-19].”

Comment 4: Page 2, 2.2. Production of hydrovoltaic generator: In this section, the most important technical details should be added.”

Response: Thank you for the suggestion, we have added the important technical details in this section:

“As the ethanol evaporates, agglomeration of the α-Al2O3 slurry begins due to increasing concentration of the α-Al2O3 nanoparticles. The capillary forces promote the aggregation upon drying of the nanoparticles. With complete drying, the nanoparticles are very close to each other and are tightly bound to the matrix, forming a porous α-Al2O3 film with abundant nanochannels.”

Comment 5: Page 3, Figure 1: This figure is too simple and should be moved into Supplementary materials.

Response: Thank you for the suggestion, we have moved Figure 1 into Supplementary materials.

Comment 6: Page 8, 4. Application: Include this section in Results and discussions.

Response: We have moved this section into “Result and discussions” as the reviewer suggested.

Reviewer 3 Report

1. Fig 2c: scale bar is not clear to see. Fig. 3f, 3g need to be improved the quality. 

2. What's the power of the device gererated ?And what's the power conversion efficencey ?

Moderate

Author Response

Comment 1: Fig 2c: scale bar is not clear to see. Fig. 3f, 3g need to be improved the quality. 

Response: Thanks for your suggestion. In the revised manuscript, the scale bar in Figure 2c was redrawn and Fig. 3f and 3g were replaced with clearer images.

Comment 2: What's the power of the device generated? And what's the power conversion efficencey ?

Response: In the revised manuscript, we have calculated the output power of the equipment.

“When the external resistance was 10 MΩ, it was equal to the internal resistance of the hydrovoltaic generator, and the maximum output power is 0.068 uW.”

In this work, it is hard to calculate the power conversion efficiency because the hydrovoltaic generator uses natural water evaporation for electricity production. We did not measure the adsorbed energy to maintain the water evaporation, and we will do it in our next work.

Round 2

Reviewer 2 Report

All my previous remarks and comments have been considered into new version of the manuscript. It means that reviewed manuscript meets the criteria and in my opinion can be published as original paper in Polymers Journal.